

# How can we reliably identify a taxon based on humeral morphology? Comparative morphology of desmostylian humeri

Kumiko Matsui

Department of Geology and Paleontology, National Museum of Nature and Science, Tsukuba, Japan

## ABSTRACT

Desmostylia is a clade of marine mammals belonging to either Tethytheria or Perissodactyla. Rich fossil records of Desmostylia were found in the Oligocene to Miocene strata of the Northern Pacific Rim, especially in the northwestern region, which includes the Japanese archipelago. Fossils in many shapes and forms, including whole or partial skeletons, skulls, teeth, and fragmentary bones have been discovered from this region. Despite the prevalent availability of fossil records, detailed taxonomic identification based on fragmentary postcranial materials has been difficult owing to to our limited knowledge of the postcranial diagnostic features of many desmostylian taxa. In this study, I propose the utilization of diagnostic characters found in the humerus to identify desmostylian genus. These characters can be used to identify isolated desmostylian humeri at the genus level, contributing to a better understanding of the stratigraphic and geographic distributions of each genus.

Corresponding author
Kumiko Matsui,
kumiko_matsui@me.com

## INTRODUCTION

Desmostylia is a clade of extinct marine mammals (*Repenning, 1965*; *Inuzuka, 1984*; *Inuzuka, 2000a*; *Inuzuka, 2000b*; *Domning, 2002*; *Gingerich, 2005*). At present, this clade is considered to belong to either Tethytheria (Afrotheria: *Domning, Ray & McKenna, 1986*) or Perissodactyla (Laurasiatheria; *Cooper et al., 2014*). Their fossil records range from the Eocene/Oligocene boundary (*Barnes & Goedert, 2001*) to the late Miocene (*Barnes, 2013*; *Barboza et al., 2017*). The last record of a definite desmostylian fossil dates from the late Miocene (*Barboza et al., 2017*). However, desmostylian remains have been found from Pliocene (*Kimura, 1966*). Many desmostylian fossils, including whole skeletons, skulls, teeth, and bones, were discovered from both the east and west sides of the North Pacific coast (*Mitchell & Repenning, 1963*; *Mitchell Jr & Lipps, 1965*; *Shikama, 1966*; *Chinzei, 1984*; *Inuzuka, 1984*; *Inuzuka, 2000a*; *Barnes & Goedert, 2001*; *Hasegawa, Kimura & Matsumoto, 2006*; *Matsui & Kawabe, 2015*).

Many diagnostic features of desmostylian genera and/or species have been proposed based on the morphology of the skull, including the mandible and molar teeth

(e.g., *Reinhart, 1959*; *Domning, Ray & McKenna, 1986*; *Inuzuka, 1988*; *Inuzuka, 2000a*; *Inuzuka, 2000b*; *Beatty, 2009*; *Chiba et al., 2016*; *Beatty & Cockburn, 2015*; *Santos, Parham & Beatty, 2016*). *Inuzuka (2000a)*, *Inuzuka (2000b)* and *Inuzuka (2013)*, for example, proposed many diagnostic features in the cranial and postcranial morphology for the genera *Desmostylus* and *Paleoparadoxia*. However, some of the proposed diagnostic features are ambiguous. There were no obvious criteria on qualitative traits. In addition, only remains of *Desmostylus* and *Paleoparadoxia* had been reported from the Miocene in Japan when his papers were published. Subsequently, another genus cf. ''*Vanderhoofius*'' sp. was described by *Chiba et al. (2016)* based on material from Hokkaido. *Santos, Parham & Beatty (2016)* provided an updated ontogenetic sequence for *Desmostylus* as well as features diagnostic of advanced age specimens based on mandibular morphology. Additionally, *Santos, Parham & Beatty (2016)* also synonymized *Vanderhoofius* with *Desmostylus*. Furthermore, *Barnes (2013)* divided the genus *Paleoparadoxia* into three genera, *Archaeoparadoxia*, *Paleoparadoxia,* and *Neoparadoxia*. His taxonomic scheme has been accepted in many studies on desmostylians (e.g., *Beatty & Cockburn, 2015*; *Matsui & Kawabe, 2015*; *Chiba et al., 2016*). Accordingly, the taxonomy of Japanese desmostylian from the Miocene needs to reflect this scheme, necessitating the establishment of diagnostic features for these three new genera. However, diagnostic features of *Paleoparadoxia* that were previously proposed by *Inuzuka (2000a)*, *Inuzuka (2000b)*, *Inuzuka (2005)* and *Inuzuka (2013)* have been applied to be specific for *Neoparadoxia* after *Barnes (2013)* split the genus into three. Therefore, postcranial diagnostic features of *Paleoparadoxia sensu stricto* have not been discussed in past studies except for those by *Shikama (1966)* and *Matsui & Kawabe (2015)*. On the other hand, there are some localities where multiple desmostylian genera were found from a single bed (e.g., Akan area; *Kimura et al., 1998*; *Sato & Kimura, 2002*; *Watanabe & Kimura, 2002*; *Yoshida & Kimura, 2002*) or similar horizons (e.g., Mizunami area, Gifu, Japan; *Yoshiwara & Iwasaki, 1902*; *Tokunaga & Iwasaki, 1914*; *Ijiri & Kamei, 1961*; *Shikama, 1966*; *Kamei & Okazaki, 1974*; *Okazaki, 1977*; *Kohno, 2000*). In such cases, it is particularly important to precisely identify desmostylian genera for recognizing their taxonomic diversity and establish detailed diagnostic characters for each genus. To rectify the current situation, a detailed comparison was made of the morphology of the humerus in the present study. As a result, diagnostic features in the humerus are proposed for each desmostylian genus.

## MATERIALS AND METHODS

### Specimens and references

In this study, I analyzed morphologies of desmostylian humeri, as well as those of potential outgroups of Desmostylia, based on direct examinations of specimens or literature reviews. The following specimens and references were used in this study (Fig. 1).

### Desmostylia
### Desmostylidae
*Ashoroa laticosta*

AMP 21, nearly complete left and right humeri of *Ashoroa laticosta* from the late Oligocene Morawan Formation, Kawakami Group, Hokkaido, Japan, described by

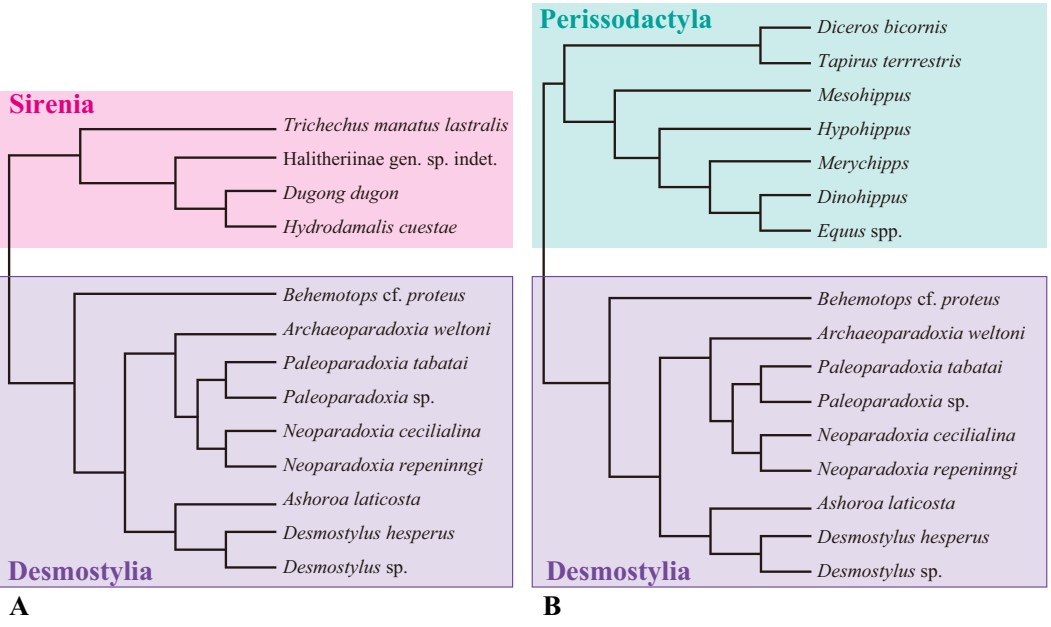

**Figure 1  Composite cladogram showing the phylogenetic relationship among taxa examined in this study.** (A) Cladgram of Desmostylia with Sirenia (Tethyteria) as an outgroup. (B) Cladgram of Perisso-dactyla as an outgroup. Compiled from numerus sources, including *Velez-Juarbe, Domning & Pyenson (2012)*, *Steiner & Ryder (2011)* and *Beatty (2009)*.

*Inuzuka (2000b)* and *Inuzuka (2011)*. This specimen is the holotype of *A. laticosta*. AMP 21 shows the epiphyseal fusion in the humerus and is considered as an adult (*Hayashi et al., 2013*; *Barnes, 2013*).

*Desmostylus hesperus*

- UHR 18466, a nearly complete left humerus of *D. hesperus* from the Middle Miocene Uchiboro coal-bearing Formation, Sakhalin, Russia. This specimen was the type specimen for *D. mirabilis* (*Nagao, 1935*), which was redescribed by *Inuzuka (1982)* and later synonymized with *D. hesperus* by Inuzuka et al. (1994). UHR 18466 shows the epiphyseal fusion in the humerus and is considered as an adult (*Hayashi et al., 2013*).

- GSJ-F7743, nearly complete left and right humeri of *D. hesperus* from the middle Miocene Tachikaraushinai Formation, Japan, described by *Inuzuka (2009)*. GSJ-F7743 does not show neurocentral fusion of vertebrae or epiphyseal fusion in long bones and is considered as a juvenile (*Hayashi et al., 2013*).

- OME-U-0170, nearly complete but proximal end was lacked, is a right humerus of *D. hesperus* from the middle Miocene Tachikaraushinai Formation, Japan. This specimen was described by *Inuzuka, Kaneko & Takabatake (2016)*. OME-U-0170 shows the epiphyseal fusion in the humerus and is considered as an adult.

*Demostylus* sp.

   *Demostylus* sp., distal part of the humerys of *Desmostylus* sp. from the Middle Miocene Chikubetsu Formation, Japan, housed in Obira City Historical Museum and reported by *Nakaya, Watabe & Akamatsu (1992)*. This specimen shows epiphyseal fusions in the humerus and is considered as an adult.

**Paleoparadoxiinae**

*Archaeoparadoxia weltoni*

   UCMP114285, incomplete and fragmentary right and left humeri of *Archaeoparadoxia weltoni* (*Clark, 1991*) from the late Oligocene or early Miocene Skooner Gulch Formation, California, USA. UCMP114285 has M3 with occlusal surface and is considered as an adult.

*Paleoparadoxia tabatai*

   NMNS PV-5601, an incomplete left humerus of *Paleoparadoxia tabatai* (*Tokunaga, 1939*) from the early Miocene Mizunami Group, Gifu, Japan, designated as the neotype of this species by *Shikama (1966)*. NMNS PV-5601 shows epiphyseal fusions in the humerus and is considered as an adult (*Hayashi et al., 2013*; *Barnes, 2013*).

*Paleoparadoxia* sp.

- SMNH VeF-61, a nearly complete left humerus of *Paleoparadoxia* sp. from the lower Miocene in the Chichibu Basin, Saitama, Japan, described by *Saegusa (2002)*. SMNH VeF-61 shows epiphyseal fusions in the humerus and is considered as an adult.
- UMUT CV31059, a proximal part of the right humerus of *Paleoparadoxia* sp. from the early Miocene Sankebetsu Formation, Hokkaido, Japan, described by *Matsui & Kawabe (2015)*. UMUT CV31059 shows epiphyseal fusions in the humerus and is considered as an adult.
- AMP AK1002, a right humerus of *Paleoparadoxia* sp. from the middle Miocene Tonokita Formation, Hokkaido, Japan. This specimen was used by *Hayashi et al. (2013)*. AMP AK1002 shows epiphyseal fusions in the humerus and is considered as an adult (*Hayashi et al., 2013*).

*Neoparadoxia cecilialina*

   LACM 150150, nearly complete right and left humeri from the lower upper Miocene Monterey Formation in California, USA. Epiphyses in humeri of LACM 150150 are not fused and the specimen is thus considered as a juvenile (*Barnes, 2013*).

*Neoparadoxia repeninngi*

   NMNS PV 20731, distal end of left humerus from the middle Miocene Ladera Formation in California, USA. Epiphyses of whole skeleton were fused and the specimen is considered as an adult.

**Family indeterminate**

*Behemotops* cf. *proteus* (*Beatty & Cockburn, 2015*)

   RBCM.EH2007.008.0001, a nearly complete left humerus from the late Oligocene of Vancouver Island, British Columbia, Canada, reported by *Beatty & Cockburn (2015)*. RBCM.EH2007.008.0001 shows epiphyseal fusions in the humerus and is considered as an adult.

*Out groups*
***Tethytheria***
Sirenia
Halithriinae gen. sp. indet.

NMNS PV-20171, a left humerus of Halitheriinae from the late Miocene Aoso Formation, Miyagi, Japan. NMNS PV-20171 shows epiphyseal fusions in the humerus and is considered as an adult.

*Hydrodamalis cuestae*

NMNS PV-21914, a cast of the right humerus of *Hydrodamalis cuestae* (SDSNH 35293; *Domning, 1978*) from the early Pleistocene San Diego Formation (Member 2), California, USA. NMNS PV-21914 shows epiphyseal fusions in the humerus and is considered as an adult.

*Dugong dugon*

NSMT M-24886, a right humerus. NSMT M-24886 shows epiphyseal fusions in the humerus and is considered as an adult.

*Trichechus manatus lastralis*

NSMT M-35016, a left humerus from USA. NSMT M-35016 shows epiphyseal fusions in the humerus and is considered as an adult.

***Perissodactyla***
Equidae (*Hermanson & MacFadden, 1992*; *Kato & Yamauchi, 2003*)

*Mesohippus*, *Merychipps*, *Hypohippus*, *Dinohippus* and *Equus* spp. illustrated in *Hermanson & MacFadden (1992)* and *Kato & Yamauchi (2003)*. All specimens are adults.

Taipiridae (*Hermanson & MacFadden, 1992*).

*Tapirus terrrestris*, illustrated in *Hermanson & MacFadden (1992)*. This is an adult specimen.

Rhinocerotidae (*Hermanson & MacFadden, 1992*)

*Diceros bicornis*, illustrated in *Hermanson & MacFadden (1992)*. This is an adult specimen.

The anatomical terminology follows *Kato & Yamauchi (2003)*. Terminologies of humorous are illustrated in Fig. 2.

## RESULTS

### Comparisons of humeral morphology between desmostylians and their outgroups

In general, the desmostylian humerus has a wide, oval, and large articular surface, as well as a large trochlea. The diaphysis of the humerus is straighter than those in Dugongidae and Trichechidae (Sirenia). It is also larger than the one in Dugongidae. The intertubercular groove is shallower and narrower in Desmostylia than in Perissodactyla. Large Perissodactyla, Equidae (larger species than *Hypohippus*) and Rhinocerotidae

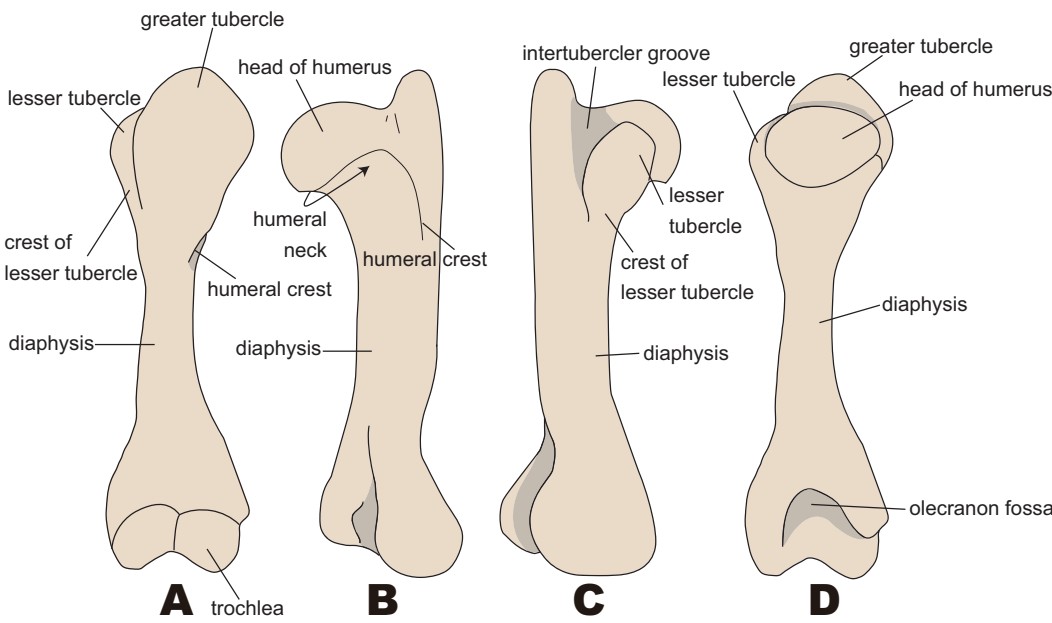

**Figure 2** **Nomenclatures of humerus (based on *Paleoparadoxia tabatai*, NMNS PV 5601, and *Paleoparadoxia* sp., UMUT CV31059).** (A) cranial side; (B) lateral side; (C) medial side; (D) caudal side.

(*Diceros bicornis*) have two intertubercular grooves and are thus very distinct from that in desmostylians. In small Perrisodactyla (Equidae smaller than *Merychippu* and Tapiridae), the greater tubercle is more developed and extended to the cranial side than in demostylians; this is the feature that clearly distinguishes this taxon from desmostylians. The humeral heads of desmostylians are oval-shaped in contrast to the semi-spherical ones in Trichechidae and *Hydrodamalis*. The lesser tubercle is developed in desmostylians, but the one in Trichechidae is fused with the greater tubercle. The greater tubercle is strongly developed and extends to the lateral side of the humerus in Dugongidae, whereas the one in desmostylians is not strongly developed on the lateral side. Additionally, dugongids have a well-developed stylate deltoid tuberosity, whereas desmostylians do not have an apparent deltoid tuberosity as do Dugongidae or Perissodactyla.

### Behemotops

The diaphysis in *Behemotops* is thinner than those in other desmostylians. The greater tubercle extends higher than the head of the humerus in *Paleoparadoxia* and *Ashoroa*. The height of this tubercle in *Behemotops* is almost the same as the one in *Ashoroa*, but smaller than the one in *Paleoparadoxia*. The curvature of the diaphysis is the greatest among desmostylians, curved along both the mediolateral side (as in *Ashoroa*) and the caudal side (as in *Trichechus* and *Hydrodamalis*). The angle of the head of the humerus is greater than those in *Ashoroa*, *Desmostylus*, *Paleoparadoxia* and is almost the same as that in *Neoparadoxia*. The intertubercular groove and lesser tubercle are not well preserved in the observed specimens of *Behemotops*. The line of attachment for the triceps muscle is not clear, unlike in *Paleoparadoxia* and *Neoparadoxia*, and is rather similar to the one in *Dugong*

*dugon.* The humeral neck of *Behemotops* is shallower than that of other desmostylians. The humeral crest is as weak as that in *Paleoparadoxia* but longer than those in *Paleoparadoxia* and *Neoparadoxia.* However, it is slightly shorter than those in *Ahoroa* and *Desmostylus.*

### Archaeoparadoxia

The preservation condition of *Archaeoparadoxia* humeri is poor, so parts available for comparison are limited. The diaphyses of the right and the left humeri are not preserved completely and thus incomparable. The humeral morphology of *Archaeoparadoxia* is similar to that of *Ashoroa* and *Paleoparadoxia* in general. The diaphyses of the right and the left humeri are curved less craniomedially than *Ashoroa* and *Behemotops,* different from *Neoparadoxia, Paleoparadoxia,* and *Desmsotylus.* The head of the humerus is oval-shaped and slightly convex at the distal end, similar to that in *Paleoparadoxia.* The lesser tubercle is distinct and medially projected, located on the medial side like *Paleoparadoxia* and different from that in *Ashoroa.* The greater tubercle is wider than that of *Behemotops* but more slender than that of *Neoparadoxia.* The lateral epicondyle is more developed and medially projected than that in *Ashoroa.* The trochlea is incomplete, smaller than that of paleoparadoxiids and desmostylids, and obliquely tilted. However, it is unknown whether the original characters are preserved in this fossil specimen.

### Neoparadoxia

The lesser and greater tubercle epiphyses are not preserved in *N. cecilialina* and *N. repeninngi,* but the direction of development and approximate size are comparable. The humeral morphology of *Neoparadoxia* is similar to that of *Paleoparadoxia* in general. The humerus of *Neoparadoxia* has a thick shaft, similar to the one found in *Paleoparadoxia.* The humeral crest is longer, extends more distally, and is more strongly developed than that in *Paleoparadoxia.* The head of the humerus is oval in shape and is horizontally longer than those in *Paleoparadoxia*, *Ashoroa,* and *Desmostylus.*

### Ashoroa

In general, the humeral morphology of *Ashoroa* is similar to that of *Paleoparadoxia* and *Archaeoparadoxia.* The lesser tubercle does not project to the medial side and is developed on the cranial side. The lesser tubercle is developed to cover the intertubercular groove and is morphologically similar to those in small-sized equids (e.g., *Mesohippus* and *Merychippus*). The humeral crest of *Ashoroa* is prominent and is developed higher and longer than in *Paleoparadoxia* and *Neoparadoxia.* It is also more robust than that in *Paleoparadoxia* and *Behemotops.*

### Desmostylus

The humeral morphology of *Desmostylus* is very different from that in other desmostylians, especially its intertubercular groove. The intertubercular groove of *Desmostylus* is located behind the head of the humerus. It is also wider and more shallow than the ones found in other desmotylians. In addition, the lesser tubercle is not knobby, unlike those in other desmostylians. The humeral crest extends distally more than the proximal half of the diaphysis and thus different from those in *Paleoparadoxia* and *Neoparadoxia.* However, it

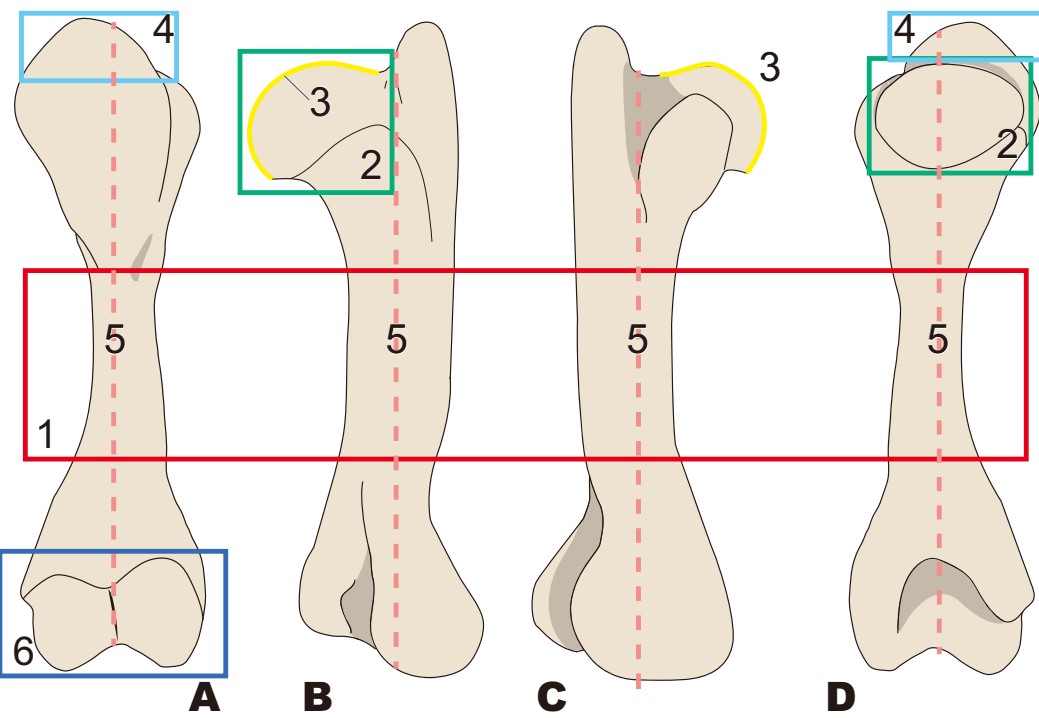

**Figure 3 Diagnostic features of Desmostylia (based on *Paleoparadoxia tabatai*, NMNS PV 5601, and *Paleoparadoxia* sp., UMUT CV31059).** The distal part is illustrated based on NMNS PV 5601, and the proximal part is illustrated based on UMUT CV31059. Numbers are corresponding to the numbers in the text. 1, Humerus diaphysis thicker than that in other relatives (red box); 2, Head of humerus larger than that in other relatives (green box); 3, Articular facet of head of humerus wider than in other relatives (yellow curve line); 4, Greater tubercle larger than other that in relatives (sky blue box); 5, Almost straight humerus diaphysis (salmon pink dotted line); 6, Trochlea larger than that in other relatives (dark blue box). (A) cranial side, (B) lateral side, (C) medial side, (D) caudal side.

appears to be similar to those in *Behemotops* and *Ashoroa*. The development of the humeral crest is greater than in *Paleoparadoxia* and *Behemotops*. The height of the greater tubercle is the same as that of the head of the humerus, differentiating it from those in *Paleoparadoxia*, *Ashoroa*, and *Behemotops*. The constriction of the diaphysis is less developed than that in *Ashoroa*, *Behemotops*, *Neoparadoxia*, and *Paleoparadoxia*.

## Diagnostic characters of desmostylian humeri

Based on the description and comparison presented above, the following combinations of diagnostic characters are proposed for each taxon.

### *Desmostylia (Fig. 3)*

1. Humerus diaphysis thicker than that in other relatives
2. Head of humerus larger than that in other relatives
3. Articular facet of head of humerus wider than in other relatives
4. Greater tubercle larger than other that in relatives
5. Almost straight humerus diaphysis
6. Trochlea larger than that in other relatives

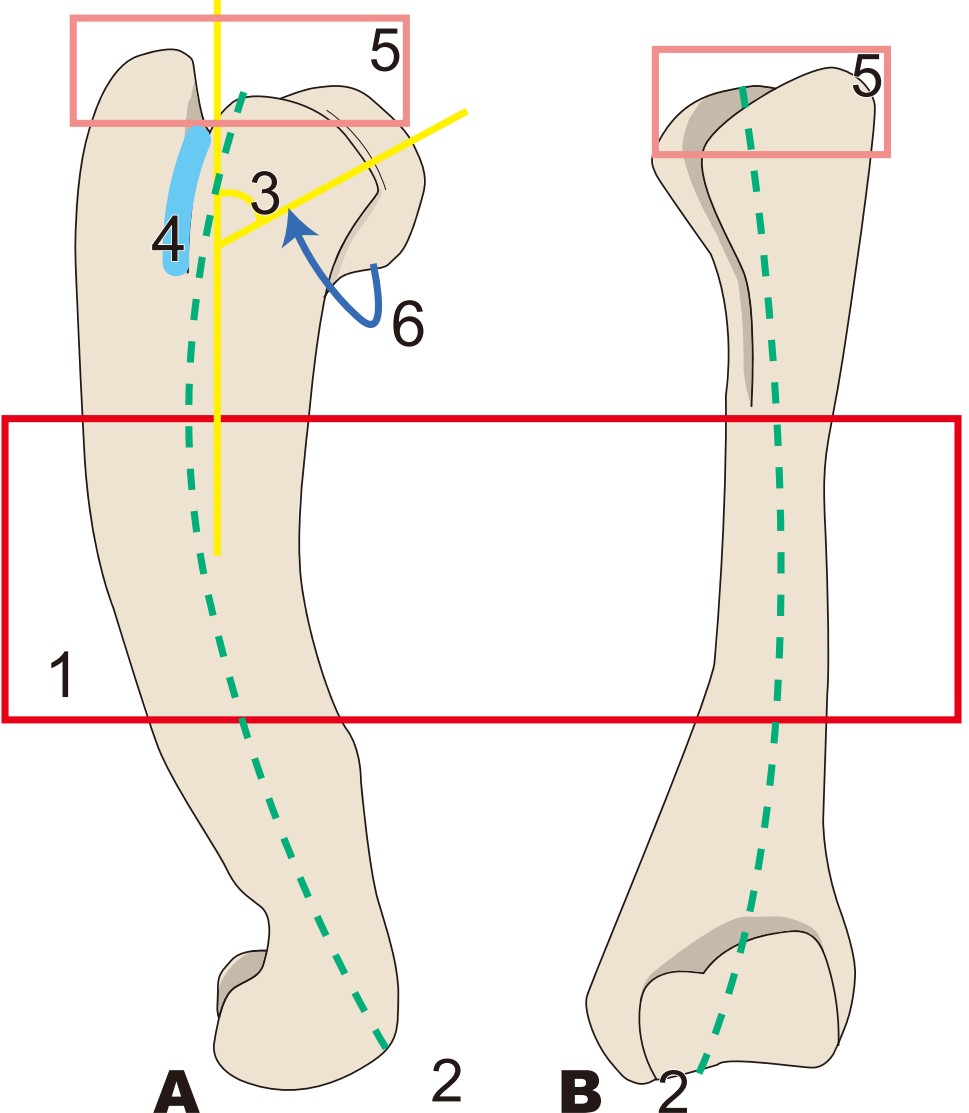

**Figure 4** **Diagnostic features of *Behemotops* (based on *Beatty & Cockburn, 2015*).** Numbers are corresponding to the numbers in the text. Humeral diaphysis thinner than that in other desmostylians (red box); 2, Diaphysis curved on both mediolateral and caudal sides as in *Trichechus* (green dot line); 3, Head of humerus with larger angle than that in other desmostylians (yellow angle); 4, Shortest intertubercular groove in desmostylians (sky blue area); 5, Greater tubercle extending dorsally higher than head of humerus (lower than that in *Paleoparadoxia*, higher than that in *Desmostylus,* and similar to that in *Ashoroa*) (salmon pink box); 6, Humeral neck shallower than that in other desmostylians (dark blue arrow line). (A) lateral side, (B) cranial side.

### *Behemotops (Fig. 4)*

1. Humeral diaphysis thinner than that in other desmostylians
2. Diaphysis curved on both mediolateral and caudal sides as in *Trichechus*
3. Head of humerus with larger angle than that in other desmostylians
4. Shortest intertubercular groove in desmostylians

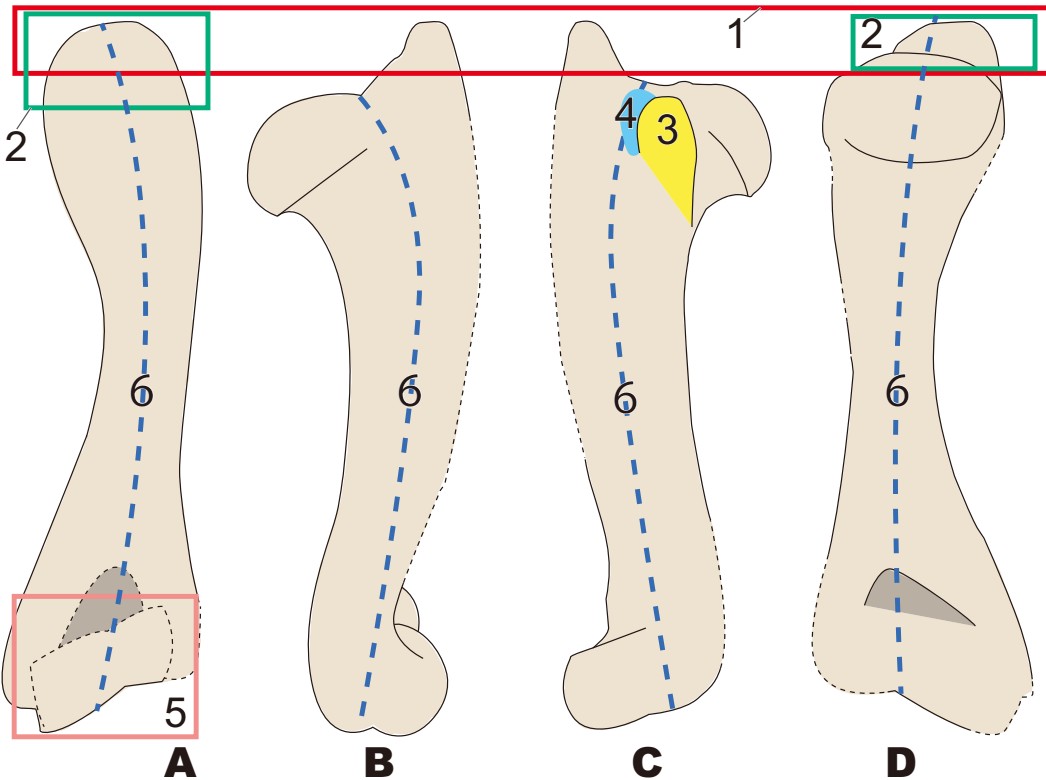

**Figure 5** **Diagnostic features of *Archaeoparadoxia* (based on UCMP114285).** Numbers are corresponding to the numbers in the text. 1, Greater tubercle extending toward proximal side above the head of the humerus as in *Paleoparadoxia* (red box); 2, Wider greater tubercle than that in *Desmostylus* and *Behemotops* (green boxes); 3, Lesser tubercle distinct and smaller than that in *Paleoparadoxia* and medially projected, located on medial side like that in *Paleoparadoxia* (yellow area); 4, Intertubercular groove located on medial side and shallower than that in *Neoparadoxia* (sky blue box); 5, Trochlea smaller than that in desmostylids and other paleoparadoxiids, but slightly larger than trochlea of *Behemotops* (dark blue circle); 7, Diaphysis slightly curved mediolaterally and caudally, unlike those of *Paleoparadoxia* and *Desmostylus,* but weaker than those of *Ashoroa* and *Behemotops* (purple boxes). (A) cranial side; (B) lateral side; (C) medial side; (D) caudal side.

5. Greater tubercle extending dorsally higher than head of humerus (lower than that in *Paleoparadoxia*, higher than that in *Desmostylus,* and similar to that in *Ashoroa*)
6. Humeral neck shallower than that in other desmostylians

### *Archaeoparadoxia (Fig. 5)*

1. Greater tubercle extending toward proximal side above the head of the humerus as in *Paleoparadoxia*
2. Wider greater tubercle than that in *Desmostylus* and *Behemotops*
3. Lesser tubercle distinct and smaller than that in *Paleoparadoxia* and medially projected, located on medial side like that in *Paleoparadoxia*
4. Intertubercular groove located on medial side and shallower than that in *Neoparadoxia*
5. Trochlea smaller than that in desmostylids and other paleoparadoxiids, but slightly larger than trochlea of *Behemotops*

 

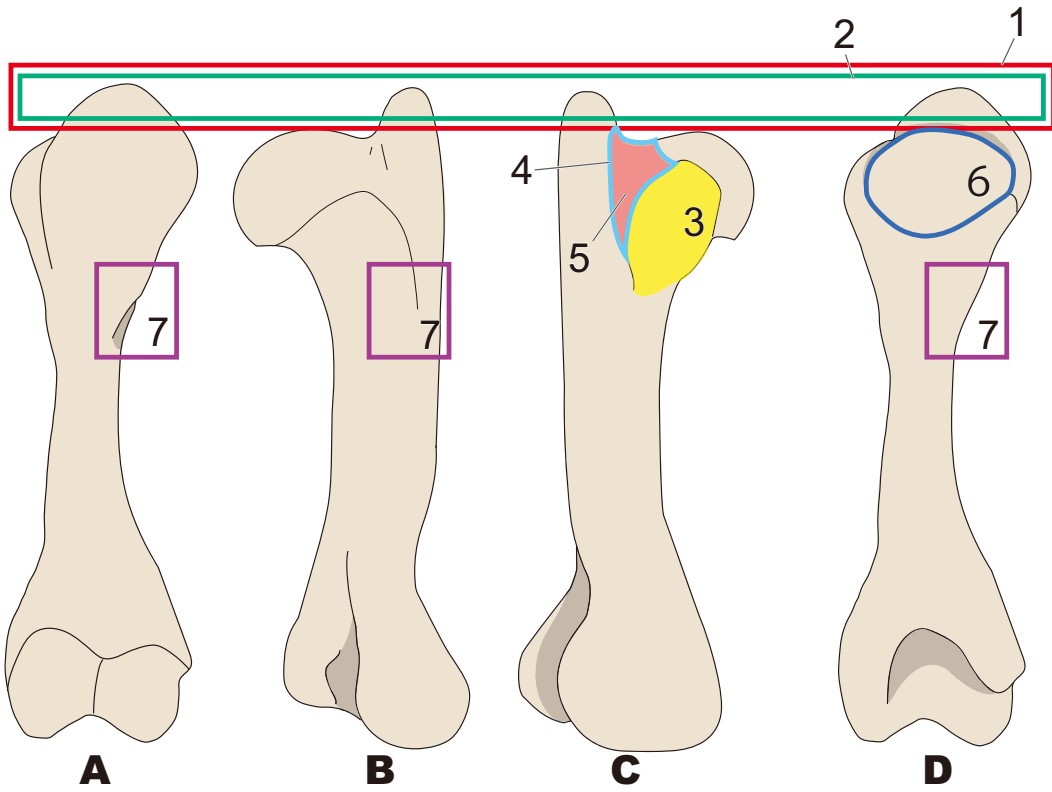

**Figure 6  Diagnostic features of *Paleoparadoxia* (based on NMNS PV 5601 and UMUT CV31059).** The distal part is illustrated based on NMNS PV 5601, and the proximal part is illustrated based on UMUT CV31059. Numbers are corresponding to the numbers in the text. 1, Greater tubercle extending toward proximal side above the head of humerus (red box); 2, Greater tubercle wider than that in *Desmostylus* and *Behemotops* (green boxes arrow line); 3, Lesser tubercle distinct and medially projected, located on medial side (yellow area); 4, Intertubercular groove located on medial side (sky blue); 5, Shallow and narrow intertubercular groove (salmon pink area); 6, Head of humerus oval-shaped and slightly convex at distal end (dark blue circle); 7, Absence of well-developed deltoid tuberosity (purple boxes). (A), cranial side; (B), lateral side; (C), medial side; (D), caudal side.

6. Diaphysis slightly curved mediolaterally and caudally, unlike those of *Paleoparadoxia* and *Desmostylus*, but weaker than those of *Ashoroa* and *Behemotops*

### *Paleoparadoxia* (*Fig. 6*; proposed by *Matsui & Kawabe, 2015*)

1. Greater tubercle extending toward proximal side above the head of the humerus
2. Greater tubercle wider than that in *Desmostylus* and *Behemotops*
3. Lesser tubercle distinct and medially projected, located on medial side
4. Intertubercular groove located on medial side
5. Shallow and narrow intertubercular groove
6. Head of humerus oval-shaped and slightly convex at distal end
7. Absence of well-developed deltoid tuberosity

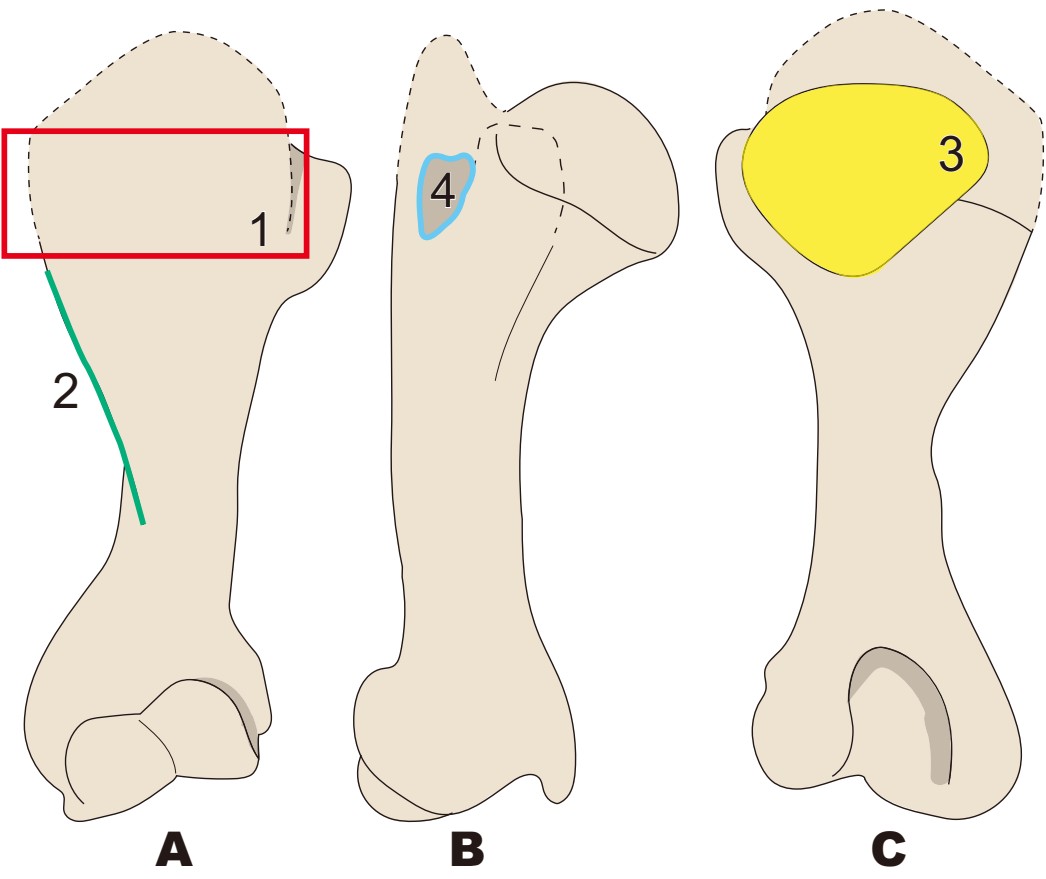

**Figure 7** **Diagnostic features of *Neoparadoxia* (based on LACM 150150 and NMNS PV 20731).** The proximal part is illustrated based on LACM 150150, and the distal part is illustrated based on NMNS PV 20731. Numbers are corresponding to the numbers in the text. 1, Greater tubercle developed as crest, stronger than that in *Paleoparadoxia* (red box); 2, Humeral crest strongly developed and extending distally over half of whole humerus (green line); 3, Head of humerus oval, wider than that in *Paleoparadoxia,* and not convex at distal end unlike in the *Paleoparadoxia* (yellow area); 4, Intertubercular groove wider than that in *Paleoparadoxia,* but narrower than that in *Desmostylus* (sky blue line). (A) cranial side; (B) lateral side; (C) caudal side.

### *Neoparadoxia* (*Fig. 7*)

1. Greater tubercle developed as crest, stronger than that in in *Paleoparadoxia*
2. Humeral crest strongly developed and extending distally over half of whole humerus
3. Head of humerus oval, wider than that in *Paleoparadoxia,* and not convex at distal end unlike in the *Paleoparadoxia*
4. Intertubercular groove wider than that in *Paleoparadoxia,* but narrower than that in *Desmostylus*

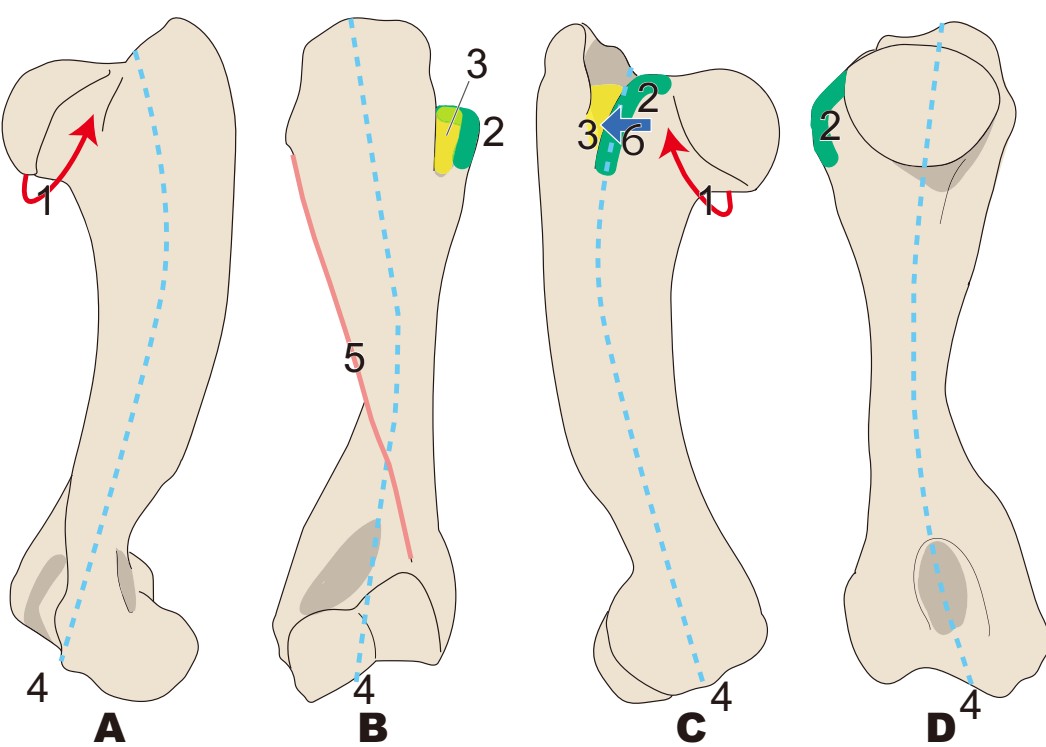

**Figure 8 Diagnostic features of *Ashoroa* (based on AMP21).** Numbers are corresponding to the numbers in the text. 1, Constriction of humeral neck shallower in desmostylians, but deeper than that in *Behemotops* (red arrow line); 2, Lesser tubercle only slightly less developed than that in *Archaeoparadoxia*, *Paleoparadoxia*, and *Neoparadoxia* (green area); 3, Intertubercular groove shorter than that in *Archaeoparadoxia*, *Paleoparadoxia*, *Neoparadoxia*, and *Desmostylus* (yellow area); 4, Diaphysis loosely curved like that in *Behemotops*, but stronger than that in *Archaeoparadoxia* (sky blue dot line); 5, Humeral crest more strongly developed than that in *Paleoparadoxia* and extending distally just above trochlea (salmon pink line); 6, Lesser tubercle located and developed on cranial side (dark blue). (A) cranial side; (B) lateral side; (C) medial side; (D) caudal side.

### *Ashoroa* (*Fig. 8*)

1. Constriction of humeral neck shallower in desmostylians, but deeper than that in *Behemotops*
2. Lesser tubercle only slightly less developed than that in *Archaeoparadoxia*, *Paleoparadoxia*, and *Neoparadoxia*
3. Intertubercular groove shorter than that in *Archaeoparadoxia*, *Paleoparadoxia*, *Neoparadoxia*, and *Desmostylus*
4. Diaphysis loosely curved like that in *Behemotops*, but stronger than that in *Archaeoparadoxia*
5. Humeral crest more strongly developed than that in *Paleoparadoxia* and extending distally just above trochlea
6. Lesser tubercle located and developed on cranial side

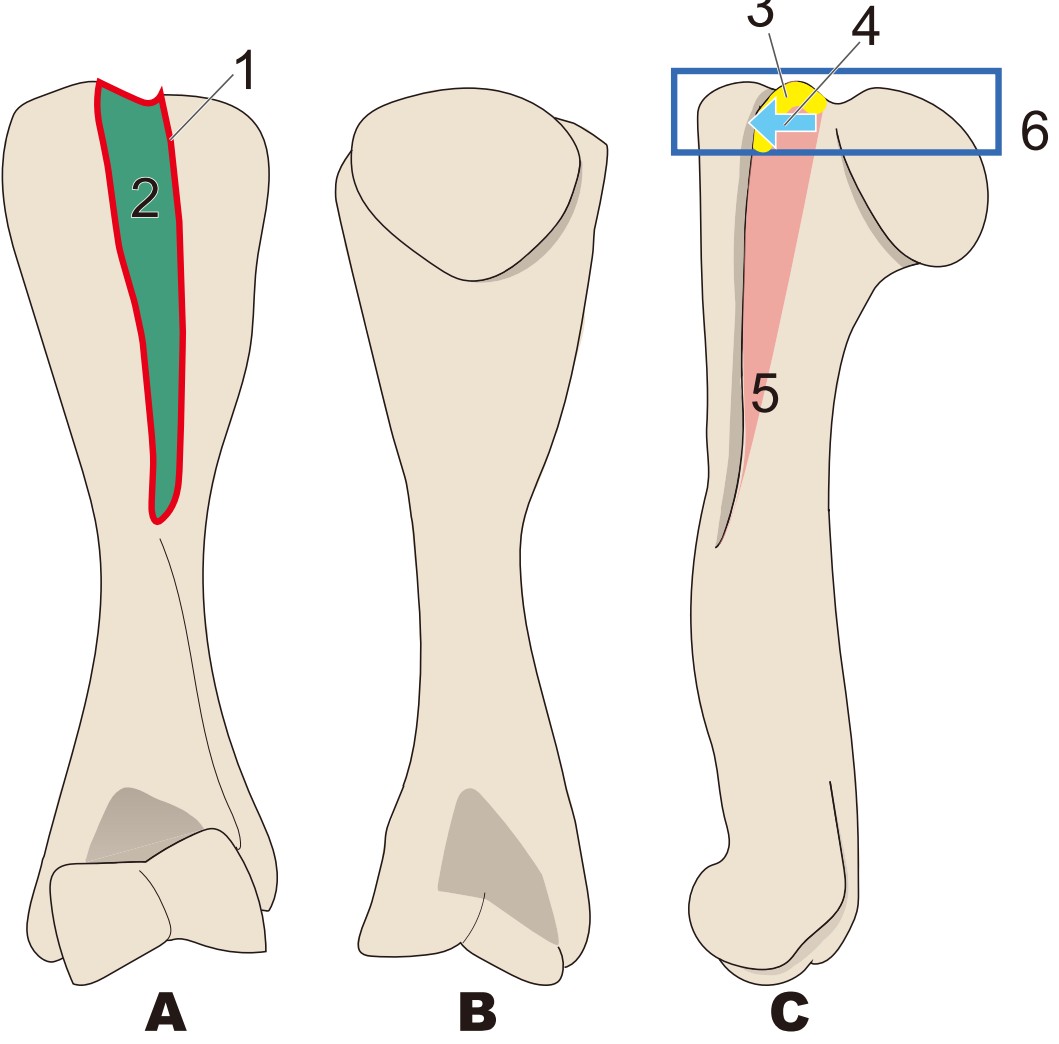

**Figure 9** **Diagnostic features of *Desmostylus* (based on UHR 18466, GSJ-F7743, and OME-U-0170).** The proximal sides of the dorsal and ventral views are illustrated based on UHR 18466, the medial side and distal part is illustrated based on UHR 18466 but has been slightly modified based on OME-U-0170 and GSJ-F7743. Numbers are corresponding to the numbers in the text. 1, Intertubercular groove located just behind head of humerus on cranial side (red circle); 2, Shallow and v-shaped intertubercular groove (green area); 3, Lesser tubercle smaller than that in other desmostylians (yellow area); 4, Lesser tubercle not projecting to medial and cranial sides (sky blue arrow line); 5, Crest of lesser tubercle well-developed and extending ventrally (salmon pink area); 6, Greater tubercle and head of humerus almost the same height (= greater tubercle not projecting higher than head of humerus) (dark blue box). (A) cranial side; (B) caudal side; (C) lateral side.

### *Desmostylus (Fig. 9)*

1. Intertubercular groove located just behind head of humerus on cranial side
2. Shallow and v-shaped intertubercular groove
3. Lesser tubercle smaller than that in other desmostylians
4. Lesser tubercle not projecting to medial and cranial sides

5.  Crest of lesser tubercle well-developed and extending ventrally
6.  Greater tubercle and head of humerus almost the same height (= greater tubercle not projecting higher than head of humerus)

## DISCUSSION

Humeral characteristics of desmostylians differ in each genus. These characters are thus sufficient for genus-level identification. The morphologies of the *Desmostylus* humerus are quite different from those in other desmostylians. The extension of the greater tubercle is shorter than that in other desmostylians. Additionally, the position of the intertubercular groove is right behind the head of humerus and very shallow compared to that in other desmostylians. These differences approximately correspond to the differences between the humeri of manatees and dugongs. Dugongs have a greater tubercle that is higher than the head of humerus and do not have an intertubercular groove that is opened right at the back of the head of the humerus, unlike manatees. The humeri of manatees show some morphological variability. Florida manatees (*Trichechus manatus*) exhibit variation in the intertubercular groove. Nineteen percent of the Florida manatees and all Amazon manatees (*Trichechus inunguis*) have an intertubercular groove, while it is absent from in other manatees (*Domning & Hayek, 1986*). The ntertubercular grooves of Amazon manatees are more distinct than those of Florida manatees (*Domning & Hayek, 1986*). These differences result from distinct biceps bracii muscles in Amazon manatees (*Domning & Hayek, 1986*). In sirenians, the hind limbs are virtually absent and locomotion is accomplished by vertical movement of the tail (*Berta, Sumich & Kovacs, 2016*). However, their locomotory use of flippers is different. Dugongs swim in the sea and use their forelimbs only for cruising (*Berta, Sumich & Kovacs, 2016*), but manatees use their forelimb to "walk" on the sea floor (*Hartman, 1979*). In Desmostylia, *Inuzuka (2013)* indicated that Paleoparadoxiinae has more movable coxae than do *Desmostylus*. However, differences in hind limbs locomotion among desmostylians have not been reported. Therefore, it has been suggested that the hind limbs of desmostylians have similar movements (*Inuzuka, 2005*). Based on fossil evidence, the humeral characteristics between *Desmostylus* and other desmostylian would likely lead to differences in swimming behavior, similar to what we observe in dugongs and manatees.

### Remaining issues

The holotype of *Desmostylus hesperus*, the type species of the genus, includes only a fragmentary molar and also does not include a humerus. Therefore, it is impossible to distinguish the proposed species of *Desmostylus* based solely on the observed diagnostic features of the holotype specimens. Accordingly, re-designating a specimen with skulls and forelimbs bearing sufficient diagnostic characters as neotypes for species of *D. hesperus* should be considered. A similar issue has been discussed for *Coelophysis bauri*, a theropod dinosaur (*Hunt & Lucas, 1991*; *Colbert et al., 1992*).

In addition, there are only six desmostylian genera, for which humeri were found in association with molars or skulls that allow us to realize taxonomic identification at the genus or species level. In other words, no postcranial skeletons are known for many

desmostylian genera or species. Accordingly, when new specimens are found in the future, the diagnostic characters proposed here would need to be evaluated and revised to reflect the new information.

## CONCLUSION

Here I present the newly established diagnostic features of desmostylian humeri. There were not many differences observed between humeral morphologies of different species of desmostylians, except for *Desmostylus*. However, these minor differences are enough to distinguish different desmostylian genera. This study will be important for taxonomic corrections and detailed classifications. Higher resolution and accurate classification than that has been previously accomplished, even for partial postcranial skeletons, would be able to achieve if new postcranial elements are identified that have highly diagnostic features. This will provide useful information for the paleogeography and distribution range of Desmostylia.

### Institutional abbreviations

| | |
|---|---|
| **AMP** | Ashoro Museum of Paleontology, Hokkaido, Japan |
| **GSJ** | Geological Survey of Japan, Ibaraki, Japan |
| **LACM** | Los Angeles County Museum, Los Angeles, USA |
| **NMNS, NSMT** | National Museum of Nature and Science, Tokyo, Japan |
| **OME** | Okhotsk Museum Esashi, 1614-1 Mikasa-cho, Esashi, Hokkaido, Japan |
| **RBCM** | Royal British Columbia Museum, Victoria, British Columbia, Canada |
| **SMNH** | Saitama Museum of Natural History, Saitama, Japan |
| **UCMP** | University of California Museum of Paleontology, Berkeley, California, USA |
| **UHR** | Hokkaido University Museum, Sapporo, Japan |
| **UMUT** | The University Museum, The University of Tokyo, Tokyo, Japan |

## ACKNOWLEDGEMENTS

I am grateful to H Naruse (Kyoto University), T Tsuihiji (The University of Tokyo), S Kawabe (FDPM) for their helpful advice. Thanks are also due to N Kohno, T Yamada, Y Tajima (NMNS), N Kaneko (GSJ), Y Kobayashi, T Tanaka (UHR), T Ando, T Sawamura, T Shinmura (AMP), O Sakamoto and H Kitagawa (SMNH), M Nagasawa (Obira Town), MB Goodwin and P Holroyd (UCMP), J Vélez-Juarbe, S McLead, VR Rhue (LACM) for access to specimens under their care. I also thank to Masamichi Ara (The University of Tokyo) for helpful advice to improve this article. Finally, thanks to the handling editor J Hutchinson, referees BL Beatty and GP Santos for constructive comments and suggestions.

### Funding

Kumiko Matsui was supported by the Japan Society for the Promotion of Science (JSPS 16J00546). The funders had no role in study design, data collection and analysis, decision to publish, or preparation of the manuscript.

### Grant Disclosures

The following grant information was disclosed by the author:
Japan Society for the Promotion of Science: 16J00546.

### Competing Interests

The author declares there are no competing interests.

### Author Contributions

- Kumiko Matsui conceived and designed the experiments, performed the experiments, analyzed the data, contributed reagents/materials/analysis tools, wrote the paper, prepared figures and/or tables, reviewed drafts of the paper.

### Data Availability

  The raw data is contained in the Results section.

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
