# Peer review of "How can we reliably identify a taxon based on humeral morphology? Comparative morphology of desmostylian humeri"

_PeerJ, doi:10.7717/peerj.4011_

## Round 0.1 · original submission · Minor Revisions

· Academic Editor

Minor Revisions

Apologies for some delay in delivering this decision but the outcome will be good news. 2 non-anonymous reviewers agree that only minor revisions are needed and I concur. They have provided detailed, constructive critiques that you will need to address point-by-point in a Response (rebuttal) document with you revised MS. Please do so and I will decide if re-review is needed.

I strongly agree that the English writing needs improvement so please undertake that- the journal does not do detailed copyediting/corrections of this kind. However the science is rigorous and that is the most important thing. Congratulations on the positive outcome, and we look forward to seeing the revised paper.

·

Basic reporting

Overall, this is a well-researched article with useful scientific information for the study of desmostylians. The use of English in this article is mostly clear and unambiguous in its presentation. Some sentences were unclear in their sentence structure or word usage. The author may benefit from having a native English speaker review the article again before submitting the final manuscript to ensure clarity to an international readership.

The section that needs the most attention is the Conclusion. This section needs to be elaborated more or simply removed entirely. One single sentence does not help to summarize the important details of this important paper for desmostylian research. Perhaps quickly summarize the differences in all desmostylian humeri and how it could help future research.

Below are sections of the article that need revision:

1. Line 35: Barboza et al., 2017 mentions evidence of Desmostylia from the early late Hemphillian Oso Member of the Capistrano Formation, which may extend the known range of Desmostylia on the eastern Pacific Coast if the material is not reworked.

2. Line 44: You may want to mention Santos et al., 2017 as the authors provide an updated ontogenetic sequence for Desmostylus as well as features diagnostic of advanced age specimens based on mandibular morphology. Additionally, the authors synonymized Vanderhoofius with Desmostylus.

3. Line 67: The additional paragraph does not seem needed. This sentence might work best being added to the end of the previous paragraph.

4. Line 75: “desmostylian” is misspelled as “desmosdtyian”

5. Line 75: This sentence is a little confusing as the author states they “examined morphology. . . based on direct examinations of specimens or literature review.” This wording is odd with the two uses of examine. Perhaps exchange “examined” with “analyzed” or something similar.

6. I would suggest moving the Institutional Abbreviation section from line 166 to the end of the Introduction or at the beginning of the Methods and Materials section. This might make it easier for the reader to have the abbreviations first instead of having to go find them below.

7. Line 138: Perhaps make it clear for readers not familiar with the outgroup relationships that the Sirenia are within Tethytheria.

8. Line 179: Depending on time for the author, creating figures the compares the stated features between the different taxa might be incredibly useful for the reader, especially those not familiar with them. I don’t believe it is absolutely necessary, but could improve the ease of understanding of this important section of the paper for the reader.

9. Line 181 and 82: Dugongidae is misspelled as Dugonidae

10. Line 181 and further mentions: Hydrodamalis is within the Dugongidae family

11. Line 185: “desmostylian” is misspelled as “desmostylidans”

12. Line 185-185: The wording within the parentheses is overly complicated and confusing to the reader. Perhaps simplifying the wording to just “Equidae smaller than Merychippus” removing the need to double parentheses.

13. Line 186-187: The line “the intertubercler groove of Perissodactyla covered by greater tubercle” may be missing a word. The sentence reads odd to me, but perhaps not.

14. Line 187: “characteristic” is misspelled as “charavteristic”

15. Line 207: “Behemotops is relatively ompared to other desmostylians.” I believe this sentence may be missing a few words.

16. Line 213: “extends to more distally” perhaps remove “to”

17. Line 229: This sentence is confusing to the reader as I am not sure what is trying to be said. “tubercle is not developed nodule-like unlike those in other desmostylians.”

18. Figures 3-7: These figures are incredibly useful and well-done, but may I suggest improvements to make the figures more useful for readers.

- On figures with multiple specimens, please label which image is which specimen for quick identification and greater use.

- Please provide explanation for the colored coded boxes and, while redundant, the characteristics associated with each number could be listed again for quick reference.

- Please make the numbering larger as some are hard to see.

- Is it possible to provide the actual specimen photos below the illustrated line drawings? This may help future researchers identify the characteristics on real world fossils as drawings may not provide enough detail that is present on the specimens.

19. Line 294: This single sentence paragraph could incorporated into the beginning of the next paragraph.

20. Line 303 and 304: This is the first mention of these Trichechus species. It may help the reader to spell out the names fully and provide taxonomic authority for the first instance.

21. Line 308: “Berta t. al” is missing the “e” in “et al.”

22. Line 310: “manatees are “walk” on the sea floor and by using their forelimb” this sentence has slightly confusing wording. Perhaps change to “manatees use their forelimb to “walk” on the sea floor” for a little more clarity.

23. Line 321: “based diagnostic features” perhaps missing “on” between “based” and “diagnostic”

24. The Conclusion section needs to be elaborated more or simply removed entirely. One single sentence does not help to summarize the important details of this important paper for desmostylian research. Perhaps quickly summarize the differences in all desmostylian humeri and how it could help future research.

25. Figure 1: Please label which groups are Desmostylia, Perissodactyla, and Tethytheria. And provide more detail on the relationship between the two others to desmostylia. The connected dotted line might be confusing to readers not familiar with desmostylian taxonomy. One solution might be to split the tree into two trees: one showing the relationship to Tethytheria and the other showing relationship to Perissodactyla.

Experimental design

Please see previous comment on suggestions for improvement to figures.

Validity of the findings

No comment.

·

Basic reporting

Aside from minor misspellings and inappropriate uses of plurals/singulars, all is in good order. See General comments to authors for further explanation.

Experimental design

The authors have put a sincere effort in justifying the use of each specimen and exploring the impacts of ontogeny on the findings. This is very good for the experimental design of the study, I have no problems with it.

Validity of the findings

The observations that differentiate the taxa are clear and supported by the evidence.

Comments for the author

The authors have submitted a manuscript outlining anatomical features that are diagnostic to the humeri of most species of desmostylians (all of those for which humeri of associated skeletons are known). Desmostylians have historically had much of their postcrania ignored in terms of taxonomic utility, and generally even though they are not all the same, many studies on locomotion tend to argue for different modes while using different taxa. The obvious thing is that those arguments could potentially all be correct, if only the authors recognized that they were all referring to different animals. Matsui and colleagues have done a great service to combat this tendency by writing to specifically identify differences among the diversity of desmostylian taxa. For this reason, I support publishing this paper.

The manuscript is not ready for publication yet, but it is only for a collection of minor revisions that are needed that are mostly related to language used. Some are about the scientific content, but others are simply copyediting matters that are not the role I should be playing here. Rather than outline every single one, here is a description of the groupings that they all fall under. It is important for the authors to go through the entire text and look for these; it will make the paper much more effective, clear, and useful:

1) Grammatical – There are some inappropriate uses of plurals and misspellings, such as “Abstracts” (should be abstract, if there is only one) and “mirabilis” (should be mirabilis). The best advice I can give is to ask a colleague to look through it to make those corrections. After seeing a manuscript so many times as an author, it is easy for one’s eyes to overlook those minor errors and fresh eyes can do a good deal of difference.

2) Clarity of terms – Throughout the text, most especially in the Results, terms such as “weakly developed”, “apparent”, “stronger”, and “robust” are used when describing morphology. Though the author may have a sense for what that means to them, these terms don’t really mean anything precise that easily translates to the same meaning to anyone else. This is a common problem in vertebrate paleontology, and something wise to avoid if you want your writing to be useful to anyone else (which is presumably why you are writing this). For example, does “desmostylians do not have an apparent deltoid tuberosity” mean something different than “desmostylians for not have a deltoid tuberosity”? All that “an apparent” does there is invoke a sense of uncertainty in your observation. If you do not observe a deltoid tuberosity, that’s unnecessary. If someone finds one later on, they can update the observation with that added detail, no harm done.
Terms like stronger/weaker/robust are meaningless is shape terms. As this is a morphological study, perhaps the better terms to use are larger/smaller/longer/shorter/wider/narrower. And, that needs to be comparative (see below).

3) Descriptions need to either be clear descriptions of shape/measurement (such as “circular”, or “5cm long”), or comparative. You have plenty of good examples of how to do that correctly here, but there are also incidences where the text includes statements of something being stronger or robust without a comparison to something else it is stronger than or more robust than. Aside from changing that to explain that the structure is larger (see point 2 above), making sure it is compared to something else is the only way to make that useful.
For example, one could say that Cornwallius has thick enamel on its molars. But thicker than what? A more useful description would state that “Cornwallius has thicker enamel on its molars than do the molars of Anthracobune, but thinner enamel than similar sized molars of Desmostylus”. If you go through the text and look for incidences in which you use a comparative term without comparing it to anything else, you should either revise it or eliminate it. The entire work will be more useful and unambiguous.

---

## Round 0.2 · Minor Revisions

· Academic Editor

Minor Revisions

Well done. Some final corrections here, please, before we can accept the MS:

line 85: "desmostylian"
lines 190-1: "the intertubercular groove of Perissodactyla developed the intertubercular groove mentioned above"-- this makes no sense. Please fix the wording so it does.
lines 217 and elsewhere: "The diaphysis of right" should say "the right" in situations like this. Please re-check.
line 221: "capti" I think you mean capitus or caput (head). It may just be better to say the head of the humerus anyway.
line 230: "epiphysis" should be "epiphyses" (plural).
line 268: "other that in relatives" should be "than in other relatives".
line 283: "Beheotops" - Behemotops
line 351: "forelimb" = "forelimbs" better

---

## Round 0.3 · Minor Revisions

· Academic Editor

Minor Revisions

The revisions introduced/revealed some more problems with the English here. I have fixed them in the annotated file (Tracked Changes) but will not fix/check further revisions - I recommend that a fluent English speaker gives this one more check to ensure the revised MS is accurate. Thank you for your patience.

---

## Round 0.4 · accepted · Accept

· Academic Editor

Accept

p18 should be headed "Discussion" not "Discussions" but otherwise the MS is acceptable- thanks!